# Amianthoid transformation of costal cartilage matrix in children with pectus excavatum and pectus carinatum

**Alexandr Kurkov**[1,2], **Anna Guller**[1,3], **Alexey Fayzullin**[1,4]*, **Nafisa Fayzullina**[5], **Vladimir Plyakin**[6], **Svetlana Kotova**[1,7], **Petr Timashev**[1,4,7,8], **Anastasia Frolova**[1], **Nikita Kurtak**[9], **Vyacheslav Paukov**[2], **Anatoly Shekhter**[1]

**1** Institute for Regenerative Medicine, Sechenov First Moscow State Medical University (Sechenov University), Moscow, Russia, **2** A.I. Strukov Department of Anatomical Pathology, Sechenov First Moscow State Medical University (Sechenov University), Moscow, Russia, **3** The Graduate School of Biomedical Engineering, University of New South Wales, Sydney, New South Wales, Australia, **4** World-Class Research Center "Digital Biodesign and Personalized Healthcare", Sechenov First Moscow State Medical University (Sechenov University), Moscow, Russia, **5** National Medical Research Center for Obstetrics, Gynecology and Perinatology Named After Academician V.I. Kulakov of the Ministry of Healthcare of the Russian Federation, Moscow, Russia, **6** Clinical and Research Institute of Emergency Pediatric Surgery and Traumatology, Moscow, Russia, **7** Department of Polymers and Composites, N.N. Semenov Institute of Chemical Physics, Moscow, Russia, **8** Chemistry Department, Lomonosov Moscow State University, Moscow, Russia, **9** FSBI "Academician V.I. Shumakov Federal Research Center of Transplantology and Artificial Organs", Ministry of Health of the Russian Federation, Moscow, Russia

* a.l.fayzullin@gmail.com

## Abstract

### Background

It is unclear if amianthoid transformation (AT) of costal cartilage extracellular matrix (ECM) has an impact on the development of pectus excavatum (PE) and pectus carinatum (PC).

### Methods

AT foci were examined in intrasurgical biopsy specimens of costal cartilages of children (8–17 years old) with PE (n = 12) and PC (n = 12) and in age-matching autopsy control samples (n = 10) using histological and immunohistochemical staining, atomic force and nonlinear optical microscopy, transmission and scanning electron microscopy, morphometry and statistics.

### Results

AT areas were identified in the costal cartilage ECM in children with normal chest, PE and PC. Each type of the AT areas ("canonical", "intertwined", "fine-fibred" and "intralacunary") had a unique morphological pattern of thickness and alignment of amianthoid fibers (AFs). AFs were formed via lateral aggregation of collagen type II fibrils in the intact ECM. Foci of the AT were observed significantly more frequently in the PE and PC groups. The AT areas had unique quantitative features in each study group.

**Data Availability Statement:** All relevant data are within the manuscript and its Supporting Information files.

**Funding:** This work was supported by the Ministry of Science and Higher education of Russian

Federation to AF and PT within the framework of state support for the creation and development of World-Class Research Centers "Digital biodesign and personalized healthcare" (N. 075-15-2020-926).

## Conclusion

AT is a structurally diverse form of ECM alteration present in healthy and pathological costal cartilage. PE and PC are associated with specific AT disorders.

## Introduction

Pectus excavatum (PE, "funnel chest") and pectus carinatum (PC, "keeled" or "pigeon" chest) are the most frequent anterior chest wall deformations that can lead to cosmetic defects, severe pain and respiratory and cardiovascular system dysfunctions [1–3]. PE is the most common (0.1–0.3% of all live births [3]) deformation manifesting as a depression of anterior chest wall at the level of $3^{rd}$-$7^{th}$ ribs. PC occurs less often (0.06% of all live births [2]) and is characterized by an outward protrusion of the sternum or the rib cage. Patients with PC have also a more favorable course of the disease. The majority of PE manifest during the first year of life [4], while PC is usually revealed in adolescence [5]. Both PE and PC are more common in males and progress over time [6].

Etiology and mechanisms of PE and PC remain unclear [2, 3, 6]. These deformations can be present as isolated defects (non-syndromic) or appear in combination with other symptoms of genetic disorders (e.g., primary connective tissue disorders) [6–8]. However, not all the PE and PC cases are congenital disorders, and some isolated forms of anterior chest deformations have a multifactorial genesis [6]. The treatment options for PE and PC include surgical and non-surgical methods. The surgical approaches vary technically [2, 3, 9–13] and one of the most common strategies includes removal of the pathologically changed costal cartilage of the deformed chest. A pathology of the costal cartilage in PE and PC patients is thought to be the cause of mechanical failure of the chest [1, 2, 6, 14, 15]. However, morphological studies of the costal cartilage tissues taken from patients with PE and PC are rare and controversial [1, 6].

The foci of amianthoid, or asbestoid, transformation (AT) is a typical finding in the extracellular matrix (ECM) of normal growing costal cartilages [16–20]. These foci appear as formations of large aggregates of thick, parallel collagen fibers (amianthoid fibers, AFs) built of laterally aggregated thin fibrils within a relatively homogenous intact matrix of the hyaline cartilage tissue. However, AFs are not localized exclusively in the costal cartilage matrix. They were also found in other hyaline cartilages (osteoarthritic articular cartilage [21], thyroid cartilage [22]), as well as in other connective tissues [23] and some tumors [24, 25]. Moreover, AFs were observed in the costal cartilages of rats [26], rabbits and pigs [20].

To date, there are no widely accepted definitions for AFs and the AT based on the morphological criteria. Histological, immunohistochemical and ultrastructural studies of AFs indicate their collagen nature [16, 17, 20, 22]. In particular, these fibers have a collagen-specific banding pattern with the periodicity of 56–67 nm [20, 24]. According to H. Claassen et al. [22], AFs in the thyroid cartilage contain collagens types II, IX and XI, resembling composition of the normal organ-specific ECM. Collagen type I, abnormal for the healthy hyaline cartilage, was found in AFs after puberty [22, 27].

The molecular mechanisms and functional significance of the AT, as well as its impact on the biomechanical properties of cartilage and pathogenesis of PE and PC still remain underexplored [17, 19, 20]. What is more, the very presence of the AT areas in PE and PC costal cartilages has been overlooked in the vast majority of morphological studies [28]. To the best of our knowledge, only one study has identified AFs in the ECM of costal cartilages of children with

PC and PE, while no such findings were observed in age-matching patients with normal chests [29].

Following this report, we conducted our own comparative study of costal cartilages of children with and without PC [30]. In contrast to the reported findings [29], we identified foci of the AT both in the healthy and PC cartilage tissues. Moreover, we classified the AT areas into four types, distinct from each other by the morphological pattern of localization within the cartilage tissue and AFs' thickness and alignment. One of these patterns, the "fine-fibred" AT area, which located in the territorial and interterritorial ECM, but not in the pericellular space, occupied a larger area in PC samples than in the normal cartilage. Furthermore, this AT area type and another two types ("canonical" and "intertwined" areas) were found in the PC cartilages significantly more often than in the control.

The present study focuses on a comparative analysis of the AT areas in the costal cartilages of children with normal chest wall, PE and PC in order to define the extent to which the AT is specific for these anterior wall chest deformations, what AT inter-type interactions and age-related changes are specific for each deformation.

## Materials and methods

### Study population

The clinical study was approved by the Local Ethical Committee of the I.M. Sechenov First Moscow State Medical University (#03–17, 19.04.2017).

Intraoperative samples of cartilage tissues were harvested from the 4th and 5th ribs of patients with PE (n = 12; 5 male, 7 female; mean age 13.7 years, ranging 8–17 years) and patients with PC (n = 12; 6 male, 6 female; mean age 13.5 years, ranging 9–17 years) during reconstructive thoracoplasty. The written informed consent for the morphological study of the collected tissue samples was obtained from the patients' legal representatives. Patients with chest traumas, pathology of thoracic viscera and musculoskeletal disorders unrelated to PE or PC were not included into this study. Autopsy fragments of the cartilages from the 4th and 5th ribs of children with a normal chest wall who died from unrelated causes (n = 10; 6 male, 4 female; mean age of 12.9 years, ranging 8–17 years) were used as control samples. The written informed consents and permissions to collect the tissue for this research were obtained from the legal representatives of these children. The total number of the studied cartilage samples was 68, including 24 biopsies in PE and PC groups and 20 autopsy samples in the control.

### Histological analysis

For the histological and immunohistochemical (IHC) studies, nonlinear optical microscopy (NLOM) and atomic force microscopy (AFM), the tissue samples were fixed in 10% neutral formalin, decalcified in an electrolytic decalcifying solution (Bio Optica, Italy), dehydrated and embedded in paraffin. The transverse 4 μm-thick microtome sections were used for histology, IHC and AFM, while 23 μm-thick sections were prepared for NLOM.

The histological staining included hematoxylin and eosin (a general overview of the tissue samples' structure), picrosirius red and Mallory's trichrome staining in Gallego's modification (for collagen), and toluidine blue (for glycosaminoglycans, GAGs). The specimens were studied with a Leica DM4000 B LED universal microscope, equipped with a Leica DFC7000 T camera running under LAS V4.8 software (Leica Microsystems, Switzerland). Two experienced pathologists performed the morphological analysis using simple, phase-contrast, dark-field and polarized light microscopy techniques.

## Nonlinear optical microscopy

For the NLOM study, deparaffinized costal cartilage sections were embedded into the «Shandon mount TM» mounting medium (USA), covered with coverslips and studied with a Zeiss LSM780 META scanning confocal microscope (Carl Zeiss, Germany). For excitation of the second harmonic generation (SHG) and two-photon fluorescence (TPF), an InSight DeepSee-OL laser (Spectra-Physics, CA, USA) with the frequency of 80 MHz and pulse length of 150 fs was used at 800 nm and 7% laser power (~2mW). Two detectors were used at 395–405 nm and 490–600 nm for SHG and TPF, respectively. The images were taken with 63× oil-immersion objective (NA = 1.4) at 1024×1024 resolution over 225 × 225 μm areas. The intensity of the SHG signal was false-color coded by the green channel brightness. TPF was registered in the second spectral channel (512–576 nm) and coded in the red color.

## Immunohistochemistry

Immunohistochemical staining for collagens types I, II and III was performed with the following primary antibodies: mouse monoclonal anti-collagen type I (GTX26308; GenTex, USA; diluted 1:4000), polyclonal rabbit anti-collagen type II (RAH C22; IMTEK, Russia; diluted 1:20) and mouse monoclonal anti-collagen type III (GTX26310; GenTex, USA; diluted 1:8000). Heat-induced antigen retrieval was performed in a citrate buffer (pH 6.0) for 15 min at 97˚C in the PT Link Rinse Station (Dako, Denmark). The sections were pretreated with 10 mg/mL hyaluronidase (NPO Microgen, RF) in Tris-buffered saline, pH 7.0 for 1 h at 37˚C. Then, secondary goat anti-mouse and goat anti-rabbit antibodies (REAL EnVision; Dako, Denmark) conjugated with horseradish peroxidase were applied and contrasted with 3,3'-diaminobenzidine chromogen (Dako, Denmark). The slides were counterstained with Mayer's hematoxylin (Sigma-Aldrich, USA). The IHC staining intensity was semi-quantitatively scored from negative (-) to triple-positive (+++) via the independent examination by two professional pathologists.

## Atomic force microscopy

AFM images were acquired from the tissue sections mounted on standard microscope slides, deparaffinized and left unstained and non-coverslipped. The imaging was performed on air in the PeakForce Tapping® mode, using a MultiMode 8 atomic force microscope with a Nanoscope V controller and E scanner (Bruker, USA). The regions of interest for AFM scanning were selected following the preliminary histological assignment and coordinated to the sample's view in the optical microscope combined with the AFM instrument. At least 5 images from different regions of each section were acquired. AFM imaging was conducted with RTE-SPA-150 and -300 probes (Bruker) with a nominal spring constant of 5 and 40 N/m, nominal frequency of 150 and 300 kHz, respectively, and a nominal tip radius of 8 nm. Detailed 5 × 5 μm images were obtained at a scan rate of 1 Hz and a 512 × 512 pixels resolution. The raw AFM images were processed using the NanoScope Analysis v.1.8 software (Bruker, USA).

## Transmission electron microscopy

For transmission electron microscopy (TEM), tissue samples were fixed in a 2.5% glutaraldehyde solution and contrasted in the saturated aqueous solution of uranyl acetate. 25 nm-thick sections were prepared with the use of a LKB-IV ultratome (LKB, Sweden), enhanced in the saturated solution of uranyl acetate and lead citrate and examined with a HT-7700 transmission electron microscope (Hitachi, Japan). The fields of view (FOVs) for TEM were preliminary selected by light microscopy examination of semi-thin 1 μm-thick tissue sections stained by the methylene blue-azure II-basic fuchsin three color method.

## Scanning electron microscopy

For scanning electron microscopy (SEM), non-fixed tissues were dried at a critical point in K 350 (Quorum, Japan), and sprayed with silver in Q150TS (Quorum, Japan). The obtained preparations were studied with a S 3400N scanning electron microscope (Hitachi, Japan).

## Morphometry

The morphometric study of the incidence and mean area of different AT types was performed via the digital image analysis. For this purpose, the costal cartilage samples were stained with picrosirius red and visualized by dark-field microscopy with a Leica DM4000 B LED universal microscope. A series of 20–50 microphotographs (the number varied according to the sample area) was taken at the magnification of 100× from each sample. Ten microphotographs were randomly selected from each series and analyzed with Adobe Photoshop CS6 software. The incidence of different AT types was determined as a percentage of AT-containing FOVs over the total number of evaluated FOVs in the sample. The mean area of each AT type was measured as a percentage of the total sample section area across the sample-specific series of microphotographs.

The thickness of collagen fibrils and their aggregates was measured by the digital analysis (Adobe Photoshop CS6 software) of TEM images taken at a magnification of 12,000×. Ten TEM images were randomly selected from the series, 15–20 per sample, for "canonical", "fine-fibred" and "intertwined" types of the AT areas and for the intact ECM. Each selected TEM image was examined in detail to evaluate the diameters of 50 selected fibrils and their aggregates. One aggregate with a maximum diameter and one fibril with a minimum diameter were chosen in each image. The remaining 48 fibrils and aggregates in the image were selected randomly. We obtained 500 values of the diameters of fibrils and aggregates, including 10 values of the minimum diameters of fibrils and 10 values of the maximal diameters of aggregates in each group, for each type of matrix in each tissue sample. The mean medium diameters, mean minimal diameters and mean maximal diameters of the fibrils and aggregates were calculated for each sample.

## Statistical analysis

The morphometry results were statistically analyzed using a two-way ANOVA followed by the Tukey's multiple comparison test in a standard program package of GraphPad Prism version 8.00 for Windows (GraphPad Software, Inc.). The results were presented as mean values with 95% confidence intervals. The correlations between the patients' age and the mean area of different types of the AT areas, and between the mean areas of different AT types were examined using the two-tailed Spearman's correlation test. The difference between the studied groups was considered statistically significant at a p value equal or less than 0.05.

# Results

## Histological analysis

The histological analysis revealed multiple AT areas in the ECM of all costal cartilage samples obtained from children with normal chests, PE and PC. These foci were observed primarily in the central zones of the cartilages and significantly differed from a visually homogenous intact matrix by the presence of clusters of numerous AFs (Figs 1–5). A specific pattern of the AFs' thickness, alignment and localization allowed us to distinguish four types of the AT areas. Each of these types was found in the costal cartilages in the normal, PE and PC groups. Different types of the AT areas gradually merged into each other (S1 Fig).

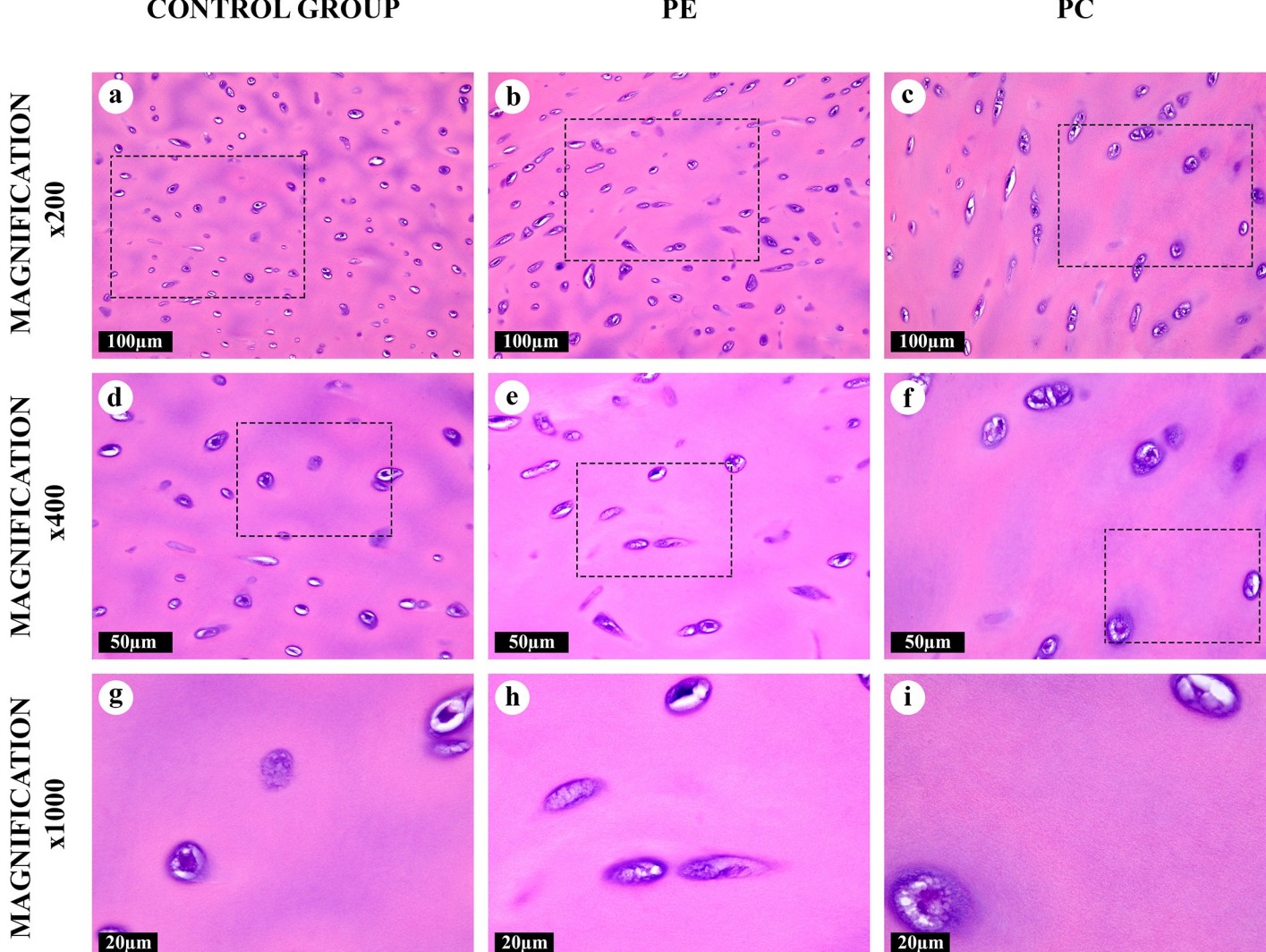

CONTROL GROUP  PE  PC

MAGNIFICATION x200

MAGNIFICATION x400

MAGNIFICATION x1000

**Fig 1. Intact matrix of the costal cartilages, hematoxylin and eosin.** Homogenous matrix with chondron structure. The territorial matrix was eosinophilic, the interterritorial matrix was weakly basophilic. The capsular (pericellular) matrix was strongly basophilic. Some chondrocytes contained vacuoles and formed isogenous cell groups.

A "canonical" type of the AT areas (Fig 2) noticeably differed from the homogeneous intact matrix which was evident even at low magnifications (×50, ×100). It was composed of thick, straight AFs arranged in parallel to each other. These fibers surrounded degenerating lacunae were observed as lens-shaped cavities (S2 Fig). Some of these fibers went through lysis and formed "puddles" (S1A Fig). The "canonical" AT areas lacked the chondron structure (normal isogenous cell groups in lacunae surrounded by pericellular, territorial and interterritorial matrix), typical for a central zone of the costal cartilage (Fig 1). "Intertwined" type of the AT areas (Fig 3) had similar morphological characteristics and was regularly merging with "canonical" type foci. However, AFs in its composition were intertwined and had a diverse braiding angle. The structural range of the "intertwined" AT areas varied from co-directional fibrils to star-shaped aggregates.

An "intralacunary" type of the AT areas (Fig 4, S2 Fig) was represented by chaotically-oriented thick bundles of the AFs inside degenerating lacunae, located near or inside the

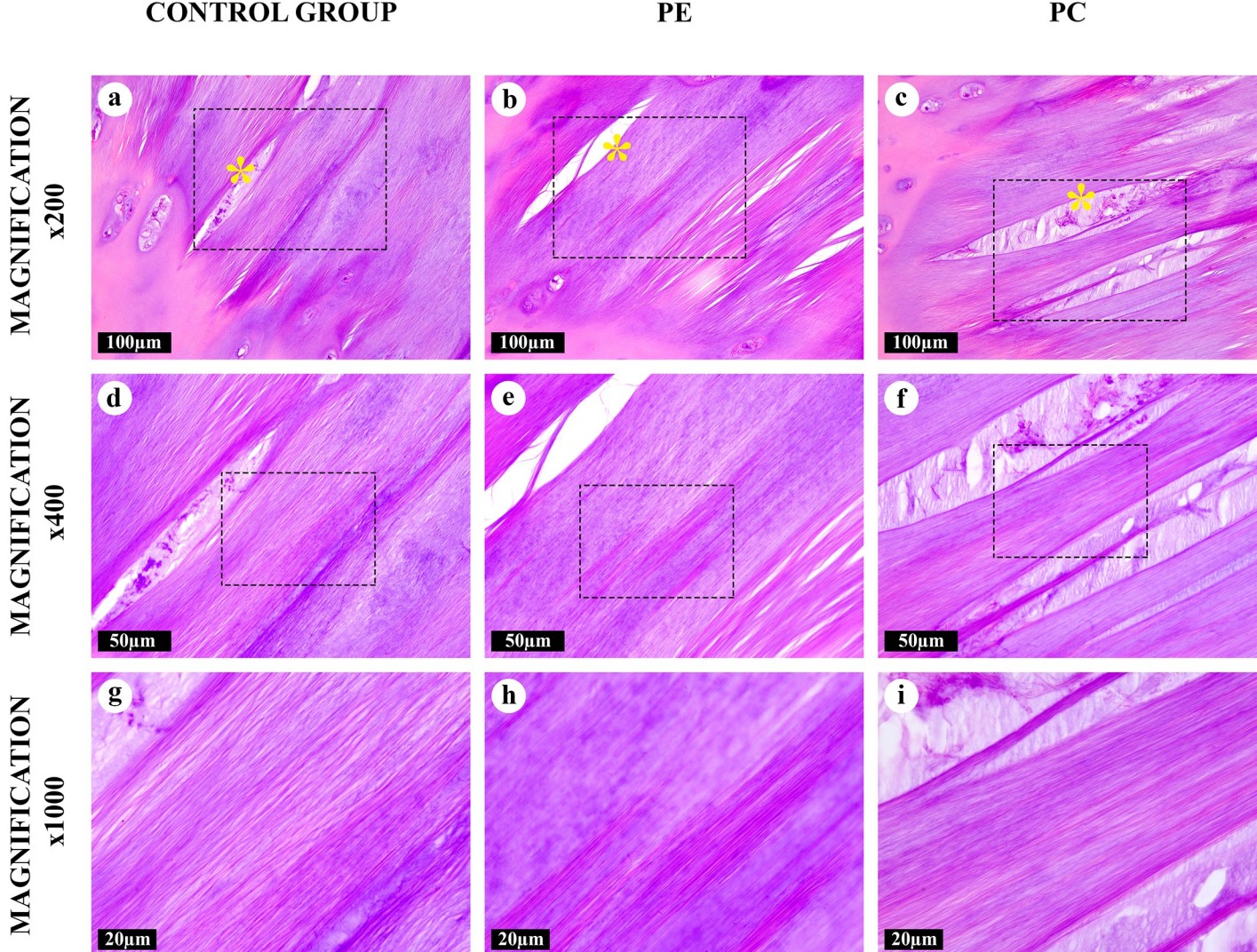

**Fig 2. "Canonical" amianthoid transformation areas in the costal cartilages, hematoxylin and eosin.** The chondron structure was absent. Aligned AFs were both eosinophilic and basophilic. Degenerating cartilaginous lacunae formed lens-shaped cavities (*) surrounded by AFs. Some cavities were filled with cellular detritus.

"canonical" and "intertwined" AT areas. This was the only type of the AT areas located inside the cartilaginous lacunae. Some aggregates of AFs in the areas of this AT type were shaped as stars. The cavities containing these AFs generally had capsules. These fibers merged with AFs in the surrounding "canonical" and "intertwined" AT areas after the capsules degenerated.

A distinctly different, "fine-fibred", type of transformed matrix (Fig 5) was represented by clusters of multidirectional thin AFs located in the territorial and predominantly in the inter-territorial matrix. The chondron structure of the hyaline cartilage tissue in these areas resembled the intact matrix (Fig 1), and the degenerated lacunae with "intralacunary" AT were absent. This AT type was commonly found in the ECM and was seen mainly at high magnifications (×400, ×600, ×1000). It was observed both independently and in a close proximity to the areas of "canonical" and "intertwined" AT types (S1 Fig).

All AFs were eosinophilic and most of them, except for AFs in the "fine-fibred" AT areas, were also basophilic when stained with hematoxylin and eosin (Figs 1–5). The picrosirius red

**CONTROL GROUP**　　　　　**PE**　　　　　**PC**

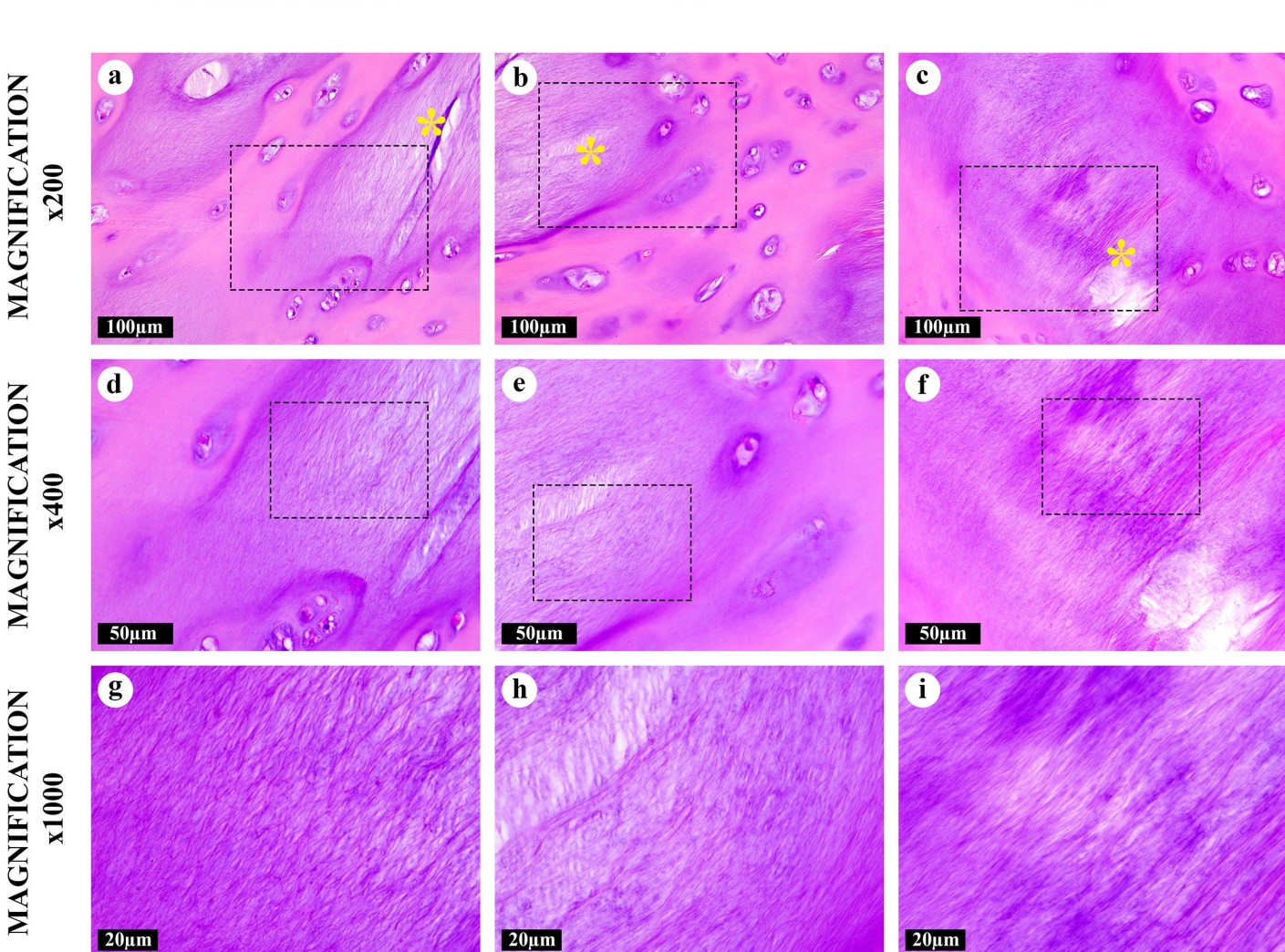

**Fig 3. "Intertwined" amianthoid transformation areas in the costal cartilages, hematoxylin and eosin.** The chondron structure was absent. AFs interlaced with each other. Lens-shaped cavities (*) surrounded by AFs.

staining of AFs gave red coloration of varying intensities, while AFs stained by Mallory were blue, with the exception of the "fine-fibred" AT areas (S3–S8 Figs) When stained with toluidine blue, metachromasia was more uneven in the "canonical", "intertwined" and "intralacunary" types of the AT areas than in the intact matrix; the "fine-fibred" AT areas were not colored with toluidine blue (S9 Fig).

The optical properties of the AT areas of all four types were similar to each other and differed significantly from the intact matrix (S3–S8 Figs). Phase-contrast microscopy revealed a distinct fibrillar structure of the matrix in all types of the AT areas, but not in the intact matrix. In dark-field and polarized light microscopy, AFs in the areas of all AT types gave a brighter glow than the intact matrix did.

The number of large lacunae with multicellular chondrocyte clones was significantly increased inside and around the areas of "canonical" and "intertwined" AT types. In these areas, chondrocytes were in the state of dystrophy (S1 and S2 Figs). Lens-shaped cavities appeared as a result of massive chondrocyte destruction in these large lacunae. They looked

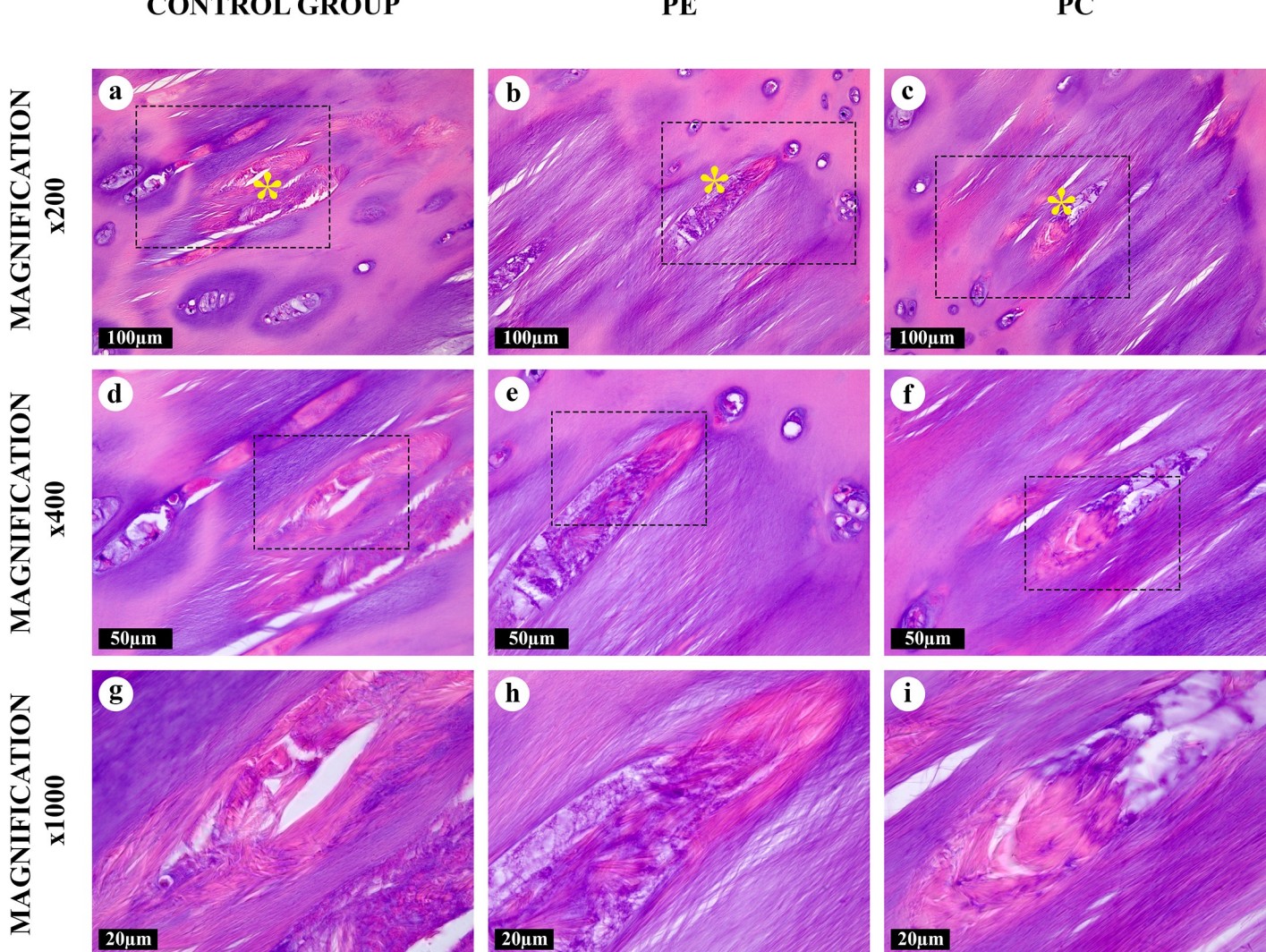

**Fig 4. "Intralacunary" amianthoid transformation areas in the costal cartilages, hematoxylin and eosin.** Some degenerating lacunae formed lens-shaped cavities containing AFs (*). The AFs were eosinophilic and oriented chaotically.

like lenticular lenses and matched the locations of damaged isogenous cell groups. After the destruction of the lacunar capsule, its contents integrated with the matrix of the "canonical" and "intertwined" AT area types. Some of these lacunae were filled with the cellular detritus or AFs of the "intralacunary" AT area type.

Some regions of the intact matrix had a weakly expressed delicate fibrillary structure due to rare thin collagen fibrils, visible by light microscopy only at high magnifications (×1000) or by phase-contrast, dark-field or polarized light microscopy (Fig 1, S3 Fig). Usually, these structures could be identified near the areas of transition into the "fine-fibred" AT areas, which made the border between them fuzzy (S1 Fig).

## Immunohistochemistry

According to the results of the IHC analysis, collagen type II dominated in all AT area types and in the intact matrix (Fig 6). Collagen types I and III, abnormal for the matrix of costal

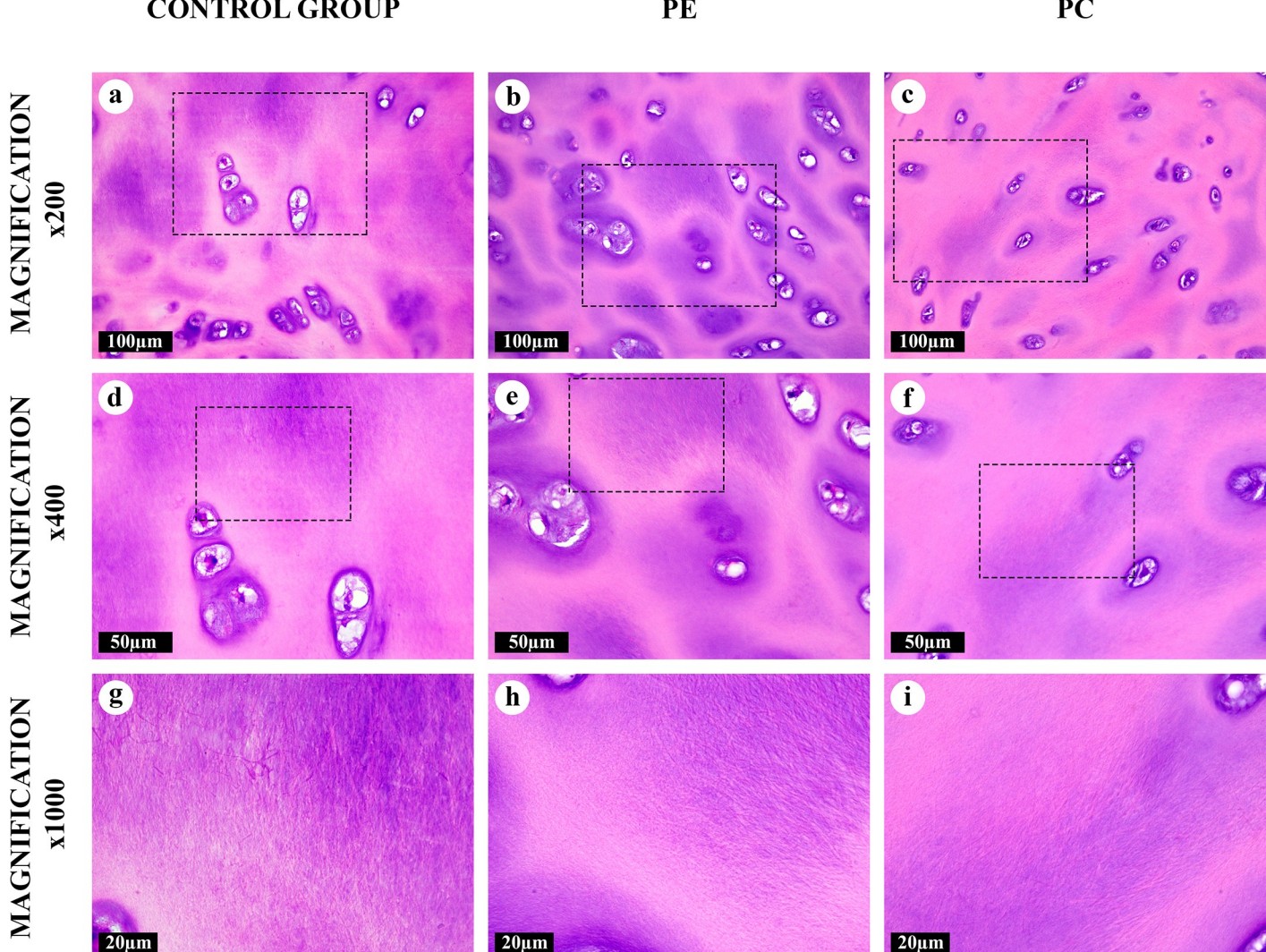

**Fig 5. "Fine-fibred" amianthoid transformation areas in the costal cartilages, hematoxylin and eosin.** The matrix was composed of the network of thin and multidirectional AFs, clearly visible only at high magnifications. The AT area had a preserved chondron structure. Tinctorial properties of territorial, inter-territorial and capsular matrices resembled the intact matrix.

cartilage, were not observed either in the intact cartilage, or in the AT areas (Table 1, S10 and S11 Figs).

## Nonlinear optical microscopy

The chondron structure, chondrocytes, pericellular (capsular), territorial and interterritorial matrix and the AT areas were identified by NLOM via variations in the intensity of SHG (green) and TPF (red) signals (Fig 7). Chondrocytes were SHG-negative and TPF-positive, while the capsular, territorial and interterritorial matrix had both SHG and TPF signals. The intact matrix was visualized as a mesh of very thin and barely visible fibers, with a diffuse SHG signal of moderate intensity. AFs in all AT area types appeared clearly distinct from the mesh fibers in the intact matrix due to their enhanced diameter and SHG emitted from AFs.

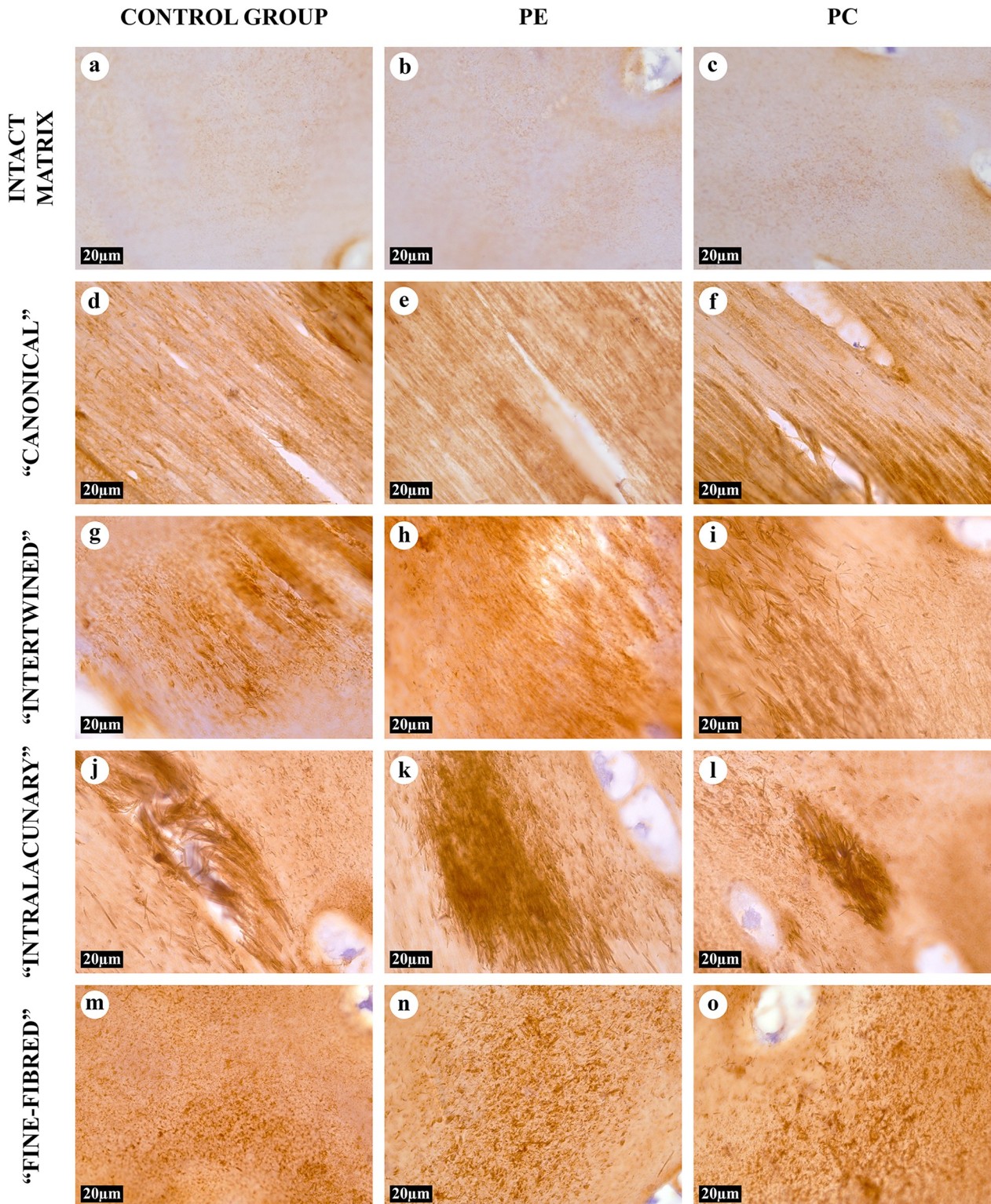

**Fig 6. IHC study of the intact matrix and four types of AT areas, anti-collagen type II antibodies, 1000×.** Collagen type II was detected in chondrocytes and all types of matrix. The expression of collagen type II in the intact matrix (A-C) was relatively weak. The AFs in the "canonical" (D-F) and "intertwined" (G-I) types of AT areas had uneven staining intensity.

**Table 1.  Immunohistochemical analysis of normal, PE and PC costal cartilages in children.**

| Costal cartilage ECM | | IHC staining | | |
|---|---|---|---|---|
| | | Collagen type I | Collagen type II | Collagen type III |
| Intact matrix | | - | From + to ++ | - |
| AFs in AT areas | "Canonical" type | - | From + to +++ | - |
| | "Intertwined" type | - | From + to +++ | - |
| | "Intralacunary" type | - | From ++ to +++ | - |
| | "Fine-fibred" type | - | From ++ to +++ | - |

Note:—- negative staining, +—weak positive staining, ++—medium positive staining, +++—high positive staining.

## Electron and atomic force microscopies

At the ultrastructural level, AFs in all types of the AT areas consisted of aggregates of collagen fibrils with a banding pattern period of 60–65 nm. The periodicity of these aggregates was equal to the periodicity of collagen fibrils of the intact matrix of costal cartilage; however, their diameter significantly exceeded the diameter of the latter. This change was caused by the observed phenomenon of sequential lateral aggregation of collagen fibrils into thicker ones, reaching a diameter of more than 500 nm. In TEM, the lateral aggregation of fibrils and transitions between different types of the AT areas were noticeably visible (Fig 8) and chondrocytes in the vicinity of the areas of "intralacunary" AT demonstrated signs of dystrophy and apoptosis.

The three-dimensional organization of all types of AT in SEM and AFM corresponded to their structure in light microscopy, NLOM and TEM (Figs 1–10).

## Morphometry

The statistical analysis of the morphometry results from TEM images showed that collagen fibrils in the "canonical" and "intertwined" AT areas were significantly thicker than fibrils of the intact matrix in all study groups (Fig 11). AFs in "fine-fibred" AT areas were significantly thicker than the fibrils of the intact matrix only in PC study group. The maximum diameters of all evaluated AFs were higher than the diameters of intact matrix fibrils in all study groups. The medium diameters of intact matrix fibrils in PE group were 47,9% higher than in the control and 21,9% higher than in PC group (Fig 12, S13 Fig). There were no other significant differences between the AF diameters of the same AT area type in the control, PE and PC groups.

The incidence of the AT areas of the "canonical", "intertwined" and "fine-fibred" types in patients with PE and PC were significantly higher than that in the control group (Fig 13A). The "fine-fibred" type was the most common AT area type in patients with PE and PC without significant differences between the groups. The "intralacunary" type was the rarest one and showed no statistical differences between the groups of children. Importantly, we discovered statistically significant differences in the incidence of AT areas between the two types of chest deformations. The "intertwined" AT areas were more frequent in the PE group than in PC, while the "canonical" AT areas were more frequently observed in patients with PC than in other groups.

The statistical analysis indicated a significant increase in the mean area of AT (all types together) in the PE and PC groups compared to the control (Fig 13B). However, no significant differences of this parameter between the studied two types of chest deformations were found. The mean area of the "fine-fibred" AT area type was significantly larger in the PE and PC groups than that in the control. This AT area type also occupied a larger area in the cartilage of

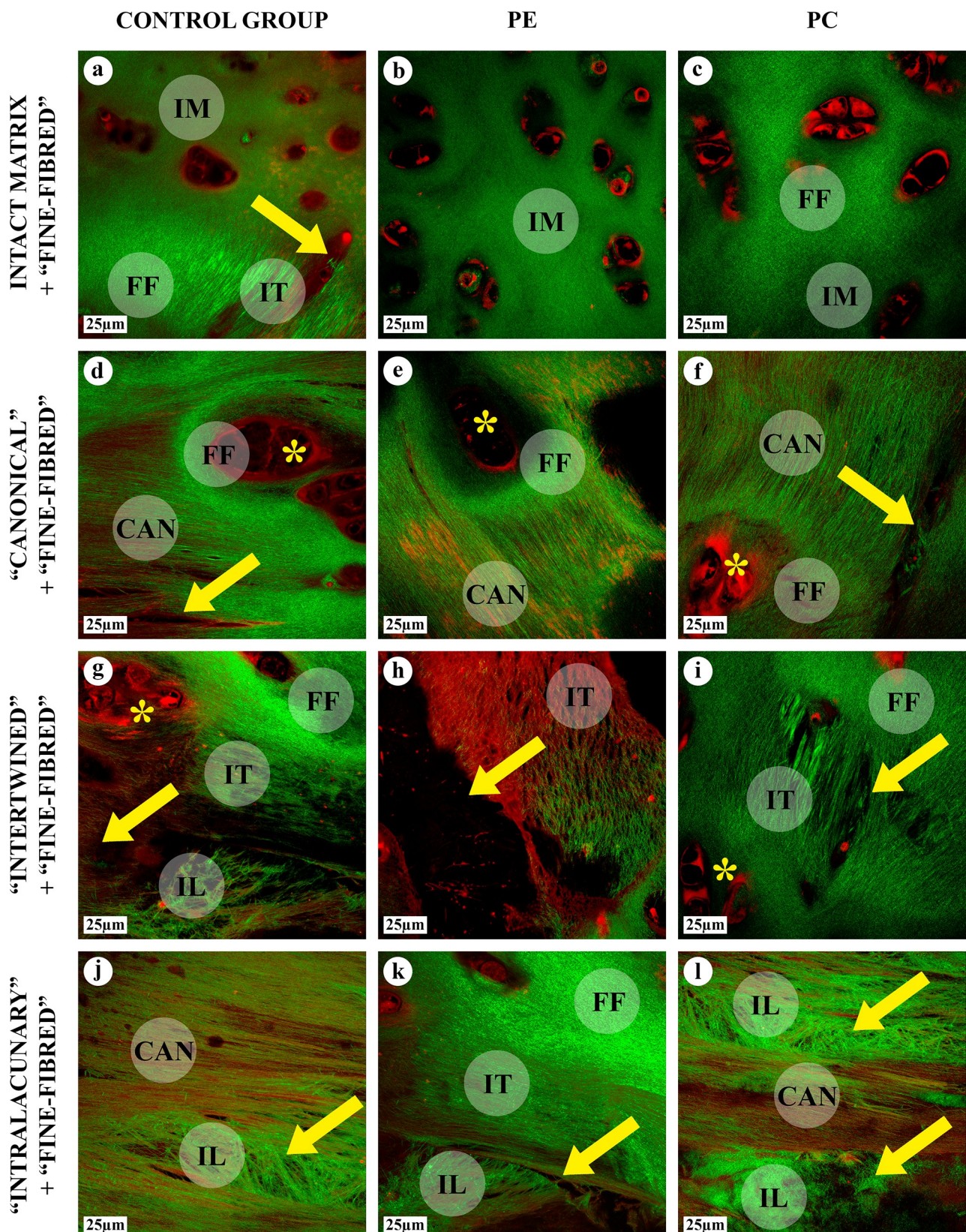

**Fig 7. NLOM of the intact matrix and four types of AT areas, SHG & TPF, ×600 (SHG–green, TPF–red).** Chondrocytes were SHG-negative and TPF-positive. Capsular, territorial and interterritorial matrix areas of matrix were predominantly SHG-positive. AFs in all types of AT areas, "canonical" (CAN), "intertwined" (IT), "intralacunary" (IL) and "fine-fibred" (FF), had an enhanced SHG-signal and a pronounced thickness. The intact matrix (IM) was observed as a delicate mesh of thin fibrils; the intact matrix and "fine-fibred" AT areas had chondron structure. Large lacunae with multicellular chondrocyte clones (*) were surrounded by a "crown" of "fine-fibred" AT area (D-F). The lacunae, lens-shaped cavities with TPF-positive cellular detritus (arrows) and "intralacunary" AFs, were often located in close proximity to "canonical" and "intertwined" AT areas. Intact matrix and different AT areas gradually merged into each other.

patients with PC than that in patients with PE, but this difference did not reach the accepted level of statistical significance. The mean area of the "intertwined" type was significantly larger in the group with PE than that in the control. In the PC group, this area was only insignificantly larger than in the control. Although, the area of the "intertwined" type considerably prevailed in the PE group compared to that in the PC group, there were no significant differences between these two groups. The areas of the "canonical" and "fine fibred" AT area types did not differ statistically in all the groups. However, in the PC group, sites of the "canonical" type tended to prevail. In all cases, the "fine-fibred" type was the most widespread AT area type, and the "intralacunary" type was the least frequent.

The correlation analysis showed differences between the control group and the groups with chest wall deformations, and between PE and PC. Positive correlations were found between the patients' age and mean areas of different AT area types in all the groups, but they tended to differ between the study groups (S1 Table).

## Discussion

In the present study, the AT areas were identified in the ECM in all the studied costal cartilage samples, both in the normal chests and in patients with PE and PC. The morphological patterns of the AT in the matrix of costal cartilages of children with PE allowed us to identify the same four types of the AT areas we had described previously in children with normal chests and PC [30]. The classification of different types of the AT foci and their detailed study allowed us to better understand the ECM structure of the normal costal cartilage, and also to reveal the specific features of the costal cartilage matrix in children with PE and PC. We discovered that, in addition to the "canonical" type, the other three AT area types also contained collagen fibrils (AFs) that were thicker than the intact matrix fibrils, proving that they are genuine and independent types of the AT areas.

The morphological and optical characteristics of AFs in all types of the AT areas identified in this study confirmed their collagenous nature and corresponded to the morphological characteristics of AFs described earlier in other studies [17, 18, 20]. The structure of each AT area type did not differ between the control and PE and PC groups. More than that, the collagen composition of AFs and the intact ECM was shown to be identical in all studied samples and consisting of collagen type II. We did not identify collagen type I in the AT areas of the costal cartilage, in contrast to the previously reported findings in the hyaline cartilage of larynxes in adults [22]. This could be attributed to the functional difference between the costal and thyroid cartilages.

The main feature of AFs is their large diameter (>120 nm, generally), significantly exceeding the diameter of collagen fibrils in the intact cartilage matrix, which is associated with the processes of lateral aggregation of fibers [17, 20]. In our study, lateral aggregation was observed in the intact cartilage matrix and in the AT areas of all types in all the study groups. The diameters of collagen fibrils differed between different types of the AT foci. The thinnest fibrils (close to the intact matrix) formed the "fine-fibred" type AT areas. On the other hand, the thickness of fibril aggregates in the AT areas of "canonical" and "intertwined" types sometimes exceeded

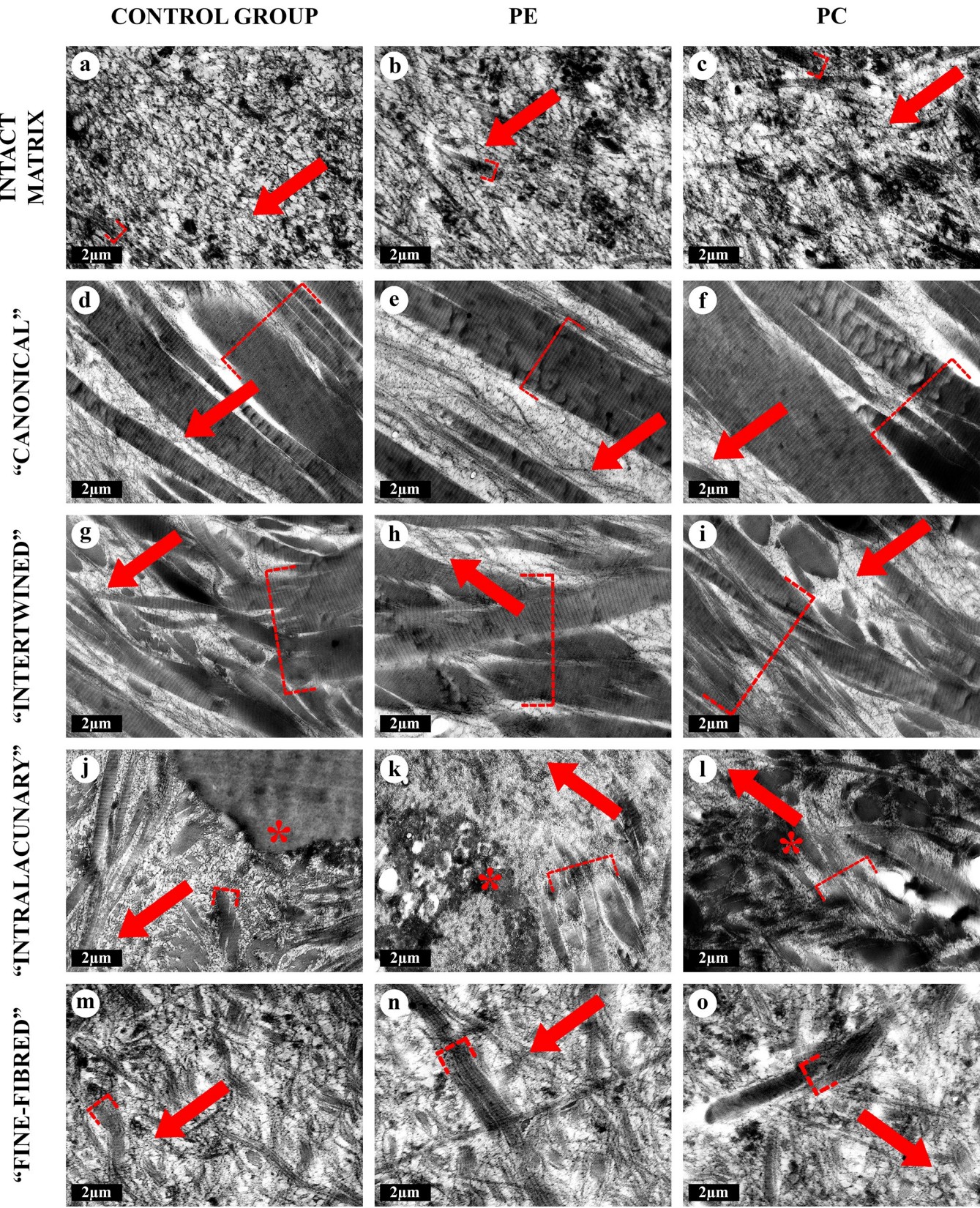

**Fig 8. TEM of the intact matrix and four types of AT areas, ×12000.** All types of matrix contained collagen fibrils of different diameters, with periodicity of 60–65 nm. The fibrils were either located separately from each other, or merged with the formation of thicker fibrils and aggregates. The intact matrix (A-C) consisted of chaotically oriented thin collagen fibrils (10–120 nm) and rare thick aggregates. In AT areas, thick fibrils (>120 nm) and their aggregates (AFs) were multiple. The AFs were aligned in "canonical" AT areas (D-F), interlacing with each other in "intertwined" AT areas (G-I) or randomly oriented in "intralacunary" (J-L) and "fine-fibred" (M-O) AT areas. The degenerating cellular debris (*) near the "intralacunary" AT area (J-L). Clusters of the thinnest fibrils are marked with arrows, the thickest aggregates were marked with dash-dotted brackets.

500 nm. However, significant differences were found only between the maximal diameters of collagen fibrils in the intact matrix and the AT areas due to the widespread presence of fibrils with a diameter of 10–120 nm.

The molecular mechanisms of lateral aggregation of collagen fibrils during the AT remain unclear. Similar processes were observed in the articular cartilage ECM, but the authors did not describe the formation of typical AFs [31]. Possibly, the lateral aggregation of collagen fibrils is associated with changes in the collagen type II synthesis in chondrocytes, post-translational modifications, and collagen folding, leading to the formation of thick collagen fibrils. It cannot be ruled out that various non-collagen molecules increasing the "stickiness" of fibrils, which were discovered by NLOM both in the intact matrix and in all types of the AT areas (TPF signals), can contribute to lateral aggregation. It was proposed that the formation of thick collagen fibrils can be associated with the formation of cross-links between collagen molecules [20] or with a specific composition of GAGs in the surrounding matrix [17]. The uneven metachromasia of "canonical", "intertwined" and "intralacunary" AT area types detected by toluidine blue staining confirmed the role of GAGs in this process. Although, the "fine-fibred" AT areas were evidently paler due to the numerous uncolored AFs. Similarly, the AFs in the "fine-fibred" AT foci did not perceive a blue dye when stained by Mallory, in contrast to the intact matrix and other types of the AT areas. However, hyaluronidase-based antigen retrieval (the enzyme degrading GAGs) was essential for the IHC without which the intact matrix and "fine-fibred" AT areas of cartilage tissue did not express any positive staining for collagen type II (S12 Fig). It can be assumed that GAGs and collagen type II molecules interact differently in AFs of the "fine-fibred" AT areas than in other regions of the ECM.

Despite their stability, AFs can be lysed by collagenases with the formation of "puddles" [20]. In our study, collagenolysis occurred in the "canonical" and "intertwined" AT areas. Unfortunately, the molecular mechanisms of this process are also underexplored.

The proliferative and dystrophic cellular reactions observed by us in the "canonical" and "intertwined" AT foci corresponded to the published data [17, 20, 32]. However, we show that AFs can accumulate inside the cartilaginous lacunae with the formation of the "intralacunar" AT areas. The relationship between the mechanisms of cellular changes and their functions with the AT process remains to be investigated.

The structural features of the AT areas probably reflect the functional significance of each type in changes of the cartilage mechanical properties of during the growth of the costal cartilages in normal conditions, as well as in chest deformities such as PE and PC. Hough and co-authors [20] suggested that the increased number of cross-links of fibrils caused the increase in the rigidity and fragility of the tissue. However, the relationship between the AT of the matrix of costal cartilage and the mechanical properties requires a detailed study. The "canonical" type most closely matches the structure of the AT areas described in the literature while the structural organizations of collagen fibers in its composition resembles tendons and ligaments [18, 22, 32]. The "intertwined" type is similar in its structure to the "canonical" one, however, it is distinguished by the arrangement of collagen fibers and fibrils and visually resembles a loose fibrous tissue. The "intralacunary" type is found only in the areas of the "canonical" and "intertwined" types, which indicates their close relationship. Collagenolysis and the cellular reactions with the destruction of lacunae occurring in the "intertwined" type let us suggest that

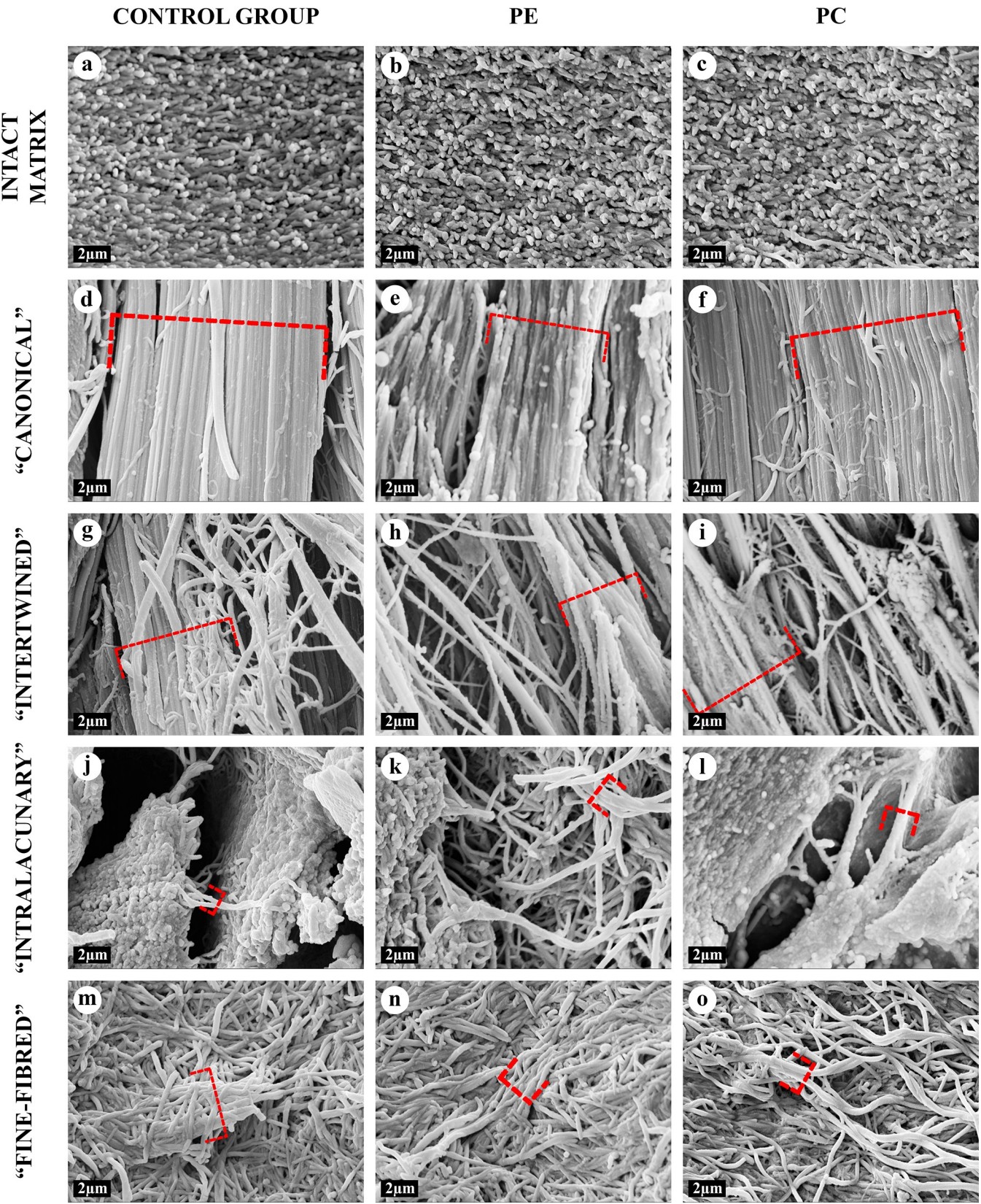

**Fig 9. SEM of the intact matrix and four types of AT areas, ×8000.** All types of matrix contained collagen fibrils of different diameters. The intact matrix (A-C) consisted of chaotically oriented thin fibrils, <120 nm in diameter. In AT areas, thick (>120 nm) fibril aggregates and AFs were multiple. The aggregates with maximal diameters were marked with dash-dotted brackets. The largest AFs were observed in "canonical" (D-F) and "intertwined" (G-I) AT areas. "Canonical" AFs were mostly oriented in parallel to each other. The "intertwined" AFs preserved the general alignment but interlaced with each other. «Intralacunary» (J-L) "fine-fibred" (M-O) AFs were multidirectional. "Intralacunary" AFs were adjoined to the surfaces of lacunae (J-L). 3D structure of "fine-fibred" AT areas resembled intact matrix.

this AT area could be a degenerative form of the ECM, so-called "amianthoid degeneration" areas. While the structure of the "canonical" AT areas indicates their role in providing the tensile strength we observed the same degenerative changes in them.

At the same time, the "fine-fibred" AT area type is the closest in its structure to the intact matrix, since the chondron structure of the hyaline cartilage tissue is preserved there. These AT areas had no signs of collagenolysis or damaged lacunae with chondrocytes, and AFs formed three-dimensional networks while having the smallest diameter among the AFs of all AT area types. It is reasonable to suggest that the formation of "fine-fibred" AT area type (with significantly thicker fibrils than those in the intact matrix) is an adaptation of the costal cartilage matrix to changes in the biomechanics of the growing chest in order to gain an additional strength.

The AT is considered an age-related change of the ECM, as the diameters of collagen matrix fibrils increase with age causing the formation of the AT areas in the second decade of life [17, 18]. In the present study, the AT foci were identified in children as young as 8 years old. However, the mean area of "canonical", "intertwined" and "intralacunary" AT foci (consisting of thick AFs) increased with age in children with normal chests.

Taken together, these findings suggest that the AT areas of all types originate from the intact ECM of the hyaline cartilage tissue via the lateral aggregation of collagen fibrils. The AT area types discovered by us in the costal cartilages represent different development stages of this process in children with normal and deformed chest. Considering the matrix structure and distribution of collagen type II fibrils, we are proposing the following hypothesis of the subsequent development of the AT (Fig 14).

Our study revealed typical qualitative features of the AT to be universal for deformed and normal chests. Both types of chest deformations demonstrated an increase in the prevalence of different types of the AT areas and total area of the AT foci. Taking into account the statistical analysis results, we can presume that the process of the AT itself was deeply altered in patients with PE and PC. Moreover, it is possible that a scarcity of cartilaginous lacunae in children with PE and PC, which we found in one of our studies [33], is associated with the excess AT. However, the morphometric characteristics of the AT areas and the correlations with patients' age were unique in each studied group.

Altogether, we demonstrated both similarities and fundamental differences in the pathological changes in the costal cartilages in children with PE and PC, which reflect the specific pathogenesis of each deformation. The causes and mechanisms of such changes have not been studied yet. However, these findings indicate the nature of the costal cartilage pathology of children with PE and PC, which could significantly contribute to the abnormalities of the cartilage tissue function. Further studies are needed to evaluate the prognostic value of the parameters of different AT area types for the improvement of treatment of PE, PC and other cartilage-related pathologies.

## Conclusion

The AT is a structurally diverse form of the interterritorial, territorial and pericellular ECM alteration present in the healthy and pathological costal cartilage. PE and PC deformations are associated with the increased incidence and mean area of the AT foci.

**CONTROL GROUP**  **PE**  **PC**

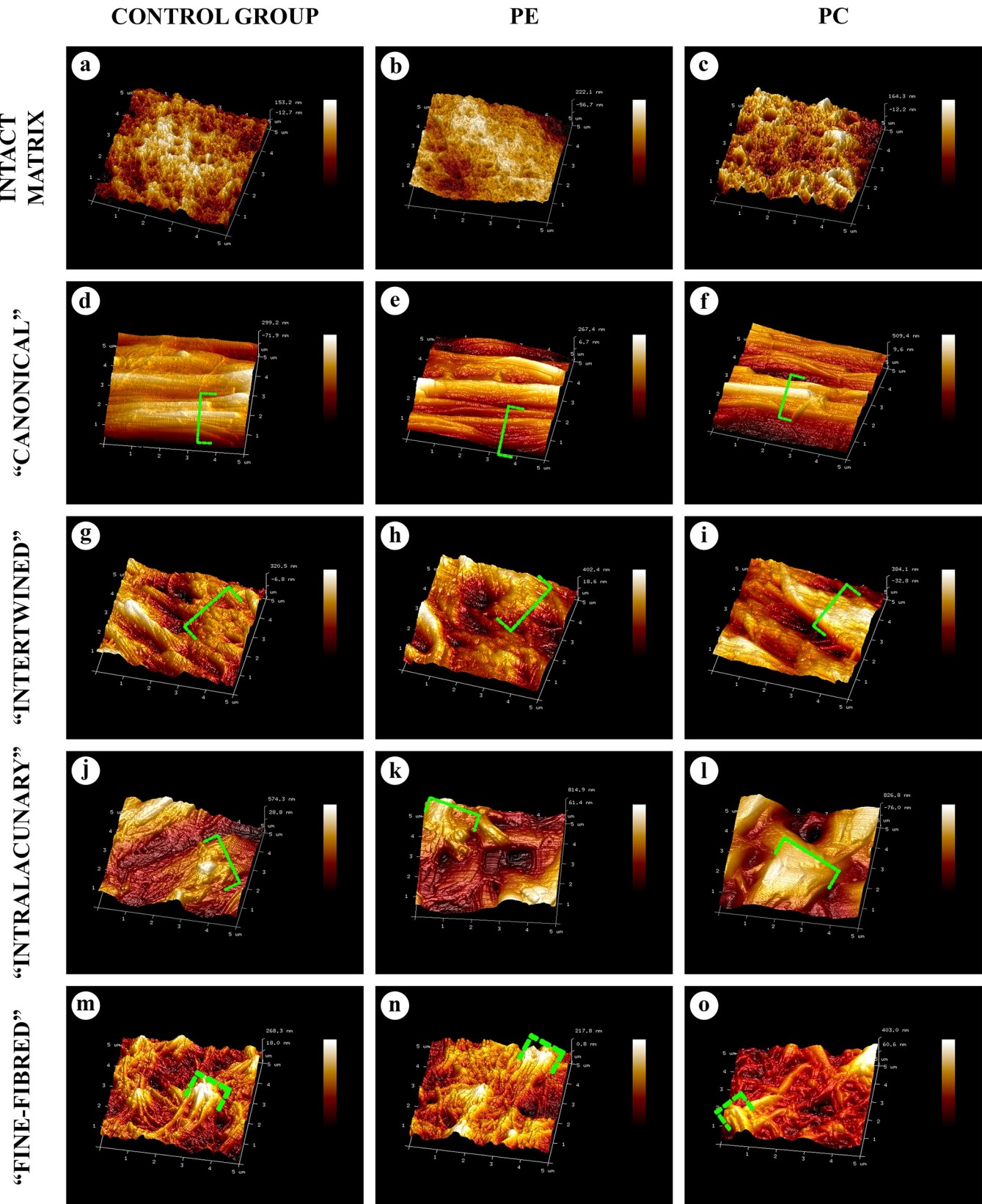

**Fig 10. AFM of the intact matrix and four types of AT areas, 5 × 5 μm.** All the matrix types contained collagen fibrils with varying diameters. The intact matrix (A-C) consisted of chaotically oriented thin (<120 nm) fibrils. In all the AT areas, there were signs of lateral aggregation of such fibrils with the formation of numerous thick (>120 nm) fibril aggregates (AFs). The 3D-structure and diameters of "canonical" AFs (D-F) distinctly differed from those of the intact matrix fibers and resembled "intertwined" AFs (G-I). The matrix in the "intralacunary" (J-L) areas consisted of thick multidirectional branching AFs. The 3D-structure of the matrix in "fine-fibred" AT areas (M-O) resembled that of the intact matrix and differed only in a larger diameter of fibrils." The "canonical" AT areas were formed by the clusters of fibrils and their aggregates oriented parallel to each other, the "intertwined"–by interlacing fibrils, and "fine-fibred" and the intact matrix represented a complex three-dimensional network. The "intralacunary" type areas were represented by chaotically located fibrils inside the cartilage lacunae, separated from the AFs of "canonical" and "intertwined" AT area types by a capsule.

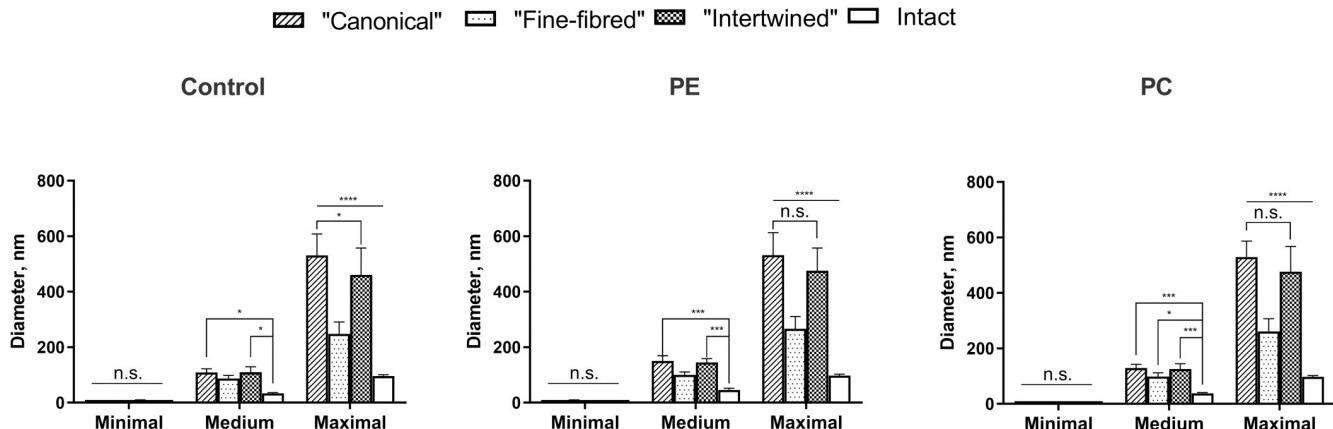

**Fig 11. Diameters of collagen fibrils in different AT areas.** Statistical significance: *—p ≤ 0.05, ***—p ≤ 0.001, ****—p ≤ 0.0001, n.s.–non-significant. Two-way ANOVA followed by the Tukey's test, mean values ± 95% confidence intervals.

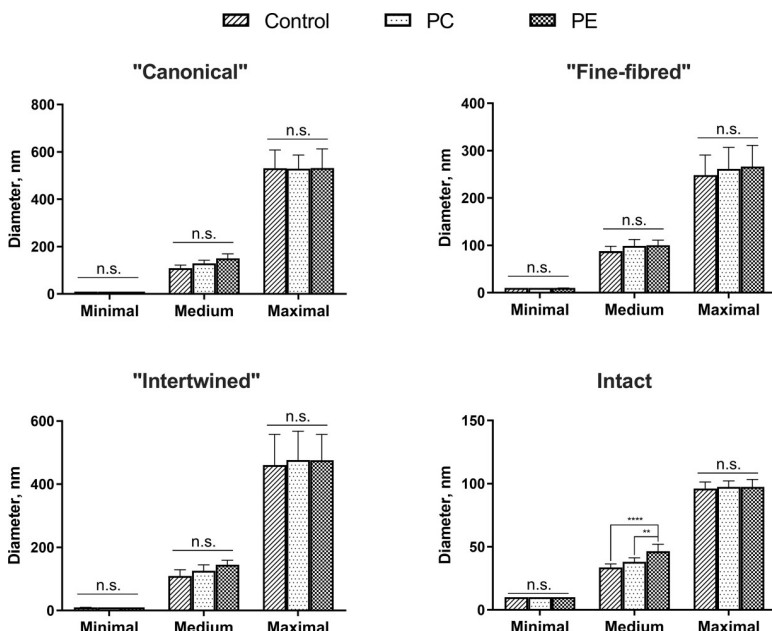

**Fig 12. Diameters of collagen fibrils in the studied groups.** Statistical significance: **—p ≤ 0.01, ****—p ≤ 0.0001, n.s.–non-significant. Two-way ANOVA followed by the Tukey's test, mean values ± 95% confidence intervals.

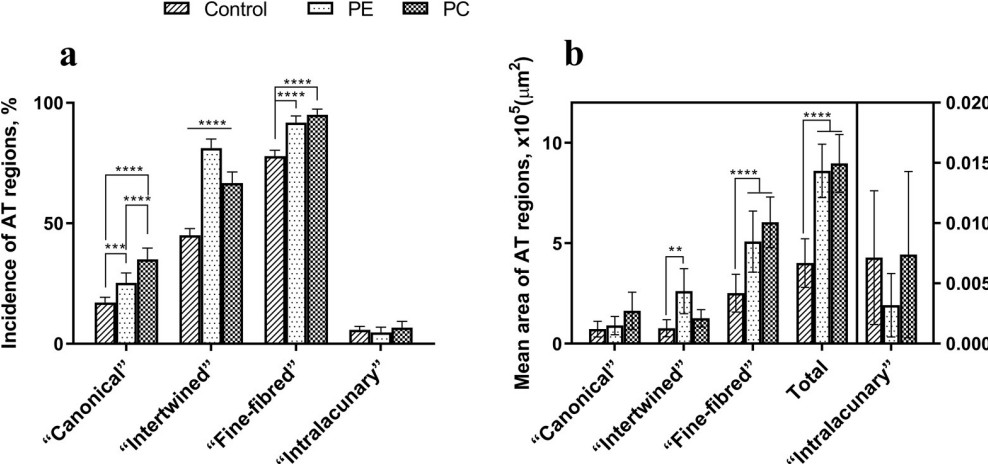

**Fig 13.** Incidence (A) and mean area (B) of AT areas in the studied groups. Statistical significance: *—p ≤ 0.05, **—p ≤ 0.01, ***—p ≤ 0.001, ****—p ≤ 0.0001. Two-way ANOVA followed by the Tukey's test, mean values ± 95% confidence intervals.

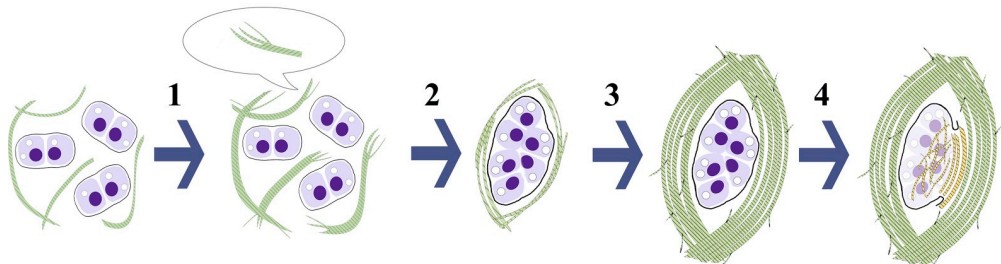

**Fig 14. Subsequent development of amianthoid transformation (AT) of costal cartilage matrix.** Collagen II fibrils (green) of the intact matrix transforms into amianthoid fibers (AFs) of "fine-fibred" AT area (1). Then, lateral aggregation of AFs leads to the formation of "canonical" or "intertwined" AT areas (2, 3) with more aligned arrangement of AFs surrounding large lacunae with dystrophic chondrocytes. Finally, the lacunae degenerate (4) with the destruction of the capsule (black border) leaving behind lens-shaped cavities with "intralacunary" AT areas (yellow) composed of collagen type II fibrils.

## Supporting information

**S1 Fig. Spatial relationships and degenerative changes in various types of AT areas in costal cartilage matrix, hematoxylin and eosin, ×1000.** Often, different types of matrix did not have clear boundaries and mixed with each other, forming intermediate, or transitional, forms. The "canonical" (CAN) and "intertwined" (IT) AT areas were separated from the intact matrix (IM) by the "fine-fibred" (FF) AT area. Upon the destruction of the capsule of degenerating cartilaginous lacunae in AT areas of the "canonical" and "intertwined" types, the contents of the lacunae merged with the surrounding matrix. "Intralacunary" (IL) AFs merged with the AFs of these two types of the AT. Amorphous basophilic contents of lacunae, cellular detritus, were observed as "puddles" in the AT areas of the "canonical" and "intertwined" types. These "puddles" could contain the foci of lysis of the "canonical", "intertwined" or "intralacunary" AFs, as well as cartilaginous lacunae. Such changes did not occur in the intact matrix or in the "fine-fibred" AT areas.
(TIF)

**S2 Fig. Cellular changes in the "canonical" and "intertwined" AT areas, hematoxylin and eosin, ×1000.** Chondrocytes were in a pronounced dystrophic state combined with hyperplasia and the formation of giant lacunae surrounded by a capsule. The destruction of chondrocytes resulted in formation of lens-shaped lacunae filled with homogeneous basophilic contents, cellular detritus, or "intralacunary" AFs.
(TIF)

**S3 Fig. Tinctorial and optical properties of the intact matrix, picrosirius red.** The intact matrix consisted of barely visible thin fibrils with anisotropic properties.
(TIF)

**S4 Fig. Tinctorial and optical properties of the "canonical" type of AT, picrosirius red.** AT areas had a distinct fibrillar structure seen by phase-contrast microscopy. The AFs were brighter than thin fibrils of the intact matrix under dark-field and polarized light microscopies.
(TIF)

**S5 Fig. Tinctorial and optical properties of the "intertwined" type of AT, picrosirius red.** AT areas had a distinct fibrillar structure seen by phase-contrast microscopy. The AFs were brighter than thin fibrils of the intact matrix under dark-field and polarized light microscopies.
(TIF)

**S6 Fig. Tinctorial and optical properties of the "intralacunary" type of AT, picrosirius red.** AT areas had a distinct fibrillar structure seen by phase-contrast microscopy. The AFs were brighter than thin fibrils of the intact matrix under dark-field and polarized light microscopies.
(TIF)

**S7 Fig. Tinctorial and optical properties of the "fine-fibred" type of AT, picrosirius red.** AT areas had a distinct fibrillar structure seen by phase-contrast microscopy. The AFs were brighter than thin fibrils of the intact matrix under dark-field and polarized light microscopies.
(TIF)

**S8 Fig.** Tinctorial properties of the intact matrix (A-C) and four types of AT areas (D-O), stained by Mallory method, ×1000. AFs in "fine-fibred" AT areas (M-O) were colorless as compared to gray-blue-purple intact matrix (A-C) and blue AFs in other types of AT (D-L). Degenerating lacunae with "intralacunary" type AF contained orange-yellow detritus (J-L).
(TIF)

**S9 Fig.** Tinctorial properties of the intact matrix (A-C) and four types of AT areas (D-O), stained with toluidine blue, ×1000. AFs in "fine-fibred" AT areas (M-O) were colorless as compared to blue-purple intact matrix (A-C). AFs in other types of AT (D-L) demonstrated uneven metachromasia.
(TIF)

**S10 Fig. IHC study of the intact matrix and four types of AT areas, anti-collagen type I antibodies, ×1000.** Collagen type I expression was absent in chondrocytes and all matrix types.
(TIF)

**S11 Fig. IHC study of the intact matrix and four types of AT areas, anti-collagen type III antibodies, ×1000.** Collagen type III expression was absent in chondrocytes and all matrix types.
(TIF)

**S12 Fig. IHC study of the intact matrix and four types of AT areas without hyaluronidase-based antigen retrieval, anti-collagen type II antibodies, ×1000.** Collagen type II expression was revealed in chondrocytes (positive internal control). Intact matrix and "fine-fibred" AT areas were not stained. The matrix of other types of AT areas had weak, uneven staining. However, most AFs did not stain positively.
(TIF)

**S13 Fig. Diameters of collagen fibrils in the studied groups.** Statistical significance: $^{**}$—$p \leq 0.01$, $^{****}$—$p \leq 0.0001$, n.s.–non-significant. Two-way ANOVA followed by the Tukey's test, interleaved scatter with bars, mean values ± 95% confidence intervals.
(TIF)

**S1 Table. Correlation analysis (only significant correlations are shown).**
(DOCX)

## Acknowledgments

Authors thank the colleagues from Laboratory for advanced studies of membrane proteins, Moscow Institute of Physics and Technology (MIPT), Dolgoprudny, Moscow Region, Russia and, personally, A.O. Bogorodskiy and V.I. Borshchevskiy.

## Author Contributions

**Conceptualization:** Alexandr Kurkov, Anna Guller, Alexey Fayzullin, Vladimir Plyakin, Petr Timashev, Nikita Kurtak, Vyacheslav Paukov, Anatoly Shekhter.

**Formal analysis:** Anna Guller, Alexey Fayzullin, Svetlana Kotova, Anatoly Shekhter.

**Investigation:** Alexandr Kurkov, Nafisa Fayzullina, Vladimir Plyakin, Svetlana Kotova, Anastasia Frolova, Nikita Kurtak, Vyacheslav Paukov, Anatoly Shekhter.

**Methodology:** Alexandr Kurkov, Alexey Fayzullin, Svetlana Kotova, Anastasia Frolova, Vyacheslav Paukov, Anatoly Shekhter.

**Project administration:** Alexandr Kurkov, Vyacheslav Paukov, Anatoly Shekhter.

**Resources:** Nafisa Fayzullina, Vladimir Plyakin, Svetlana Kotova, Petr Timashev, Nikita Kurtak, Anatoly Shekhter.

**Supervision:** Anna Guller, Svetlana Kotova, Petr Timashev.

**Visualization:** Anna Guller, Nafisa Fayzullina, Anastasia Frolova, Anatoly Shekhter.

**Writing – original draft:** Alexandr Kurkov, Alexey Fayzullin, Anatoly Shekhter.

**Writing – review & editing:** Anna Guller, Nafisa Fayzullina, Vladimir Plyakin, Svetlana Kotova, Petr Timashev, Anastasia Frolova, Nikita Kurtak, Vyacheslav Paukov.

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
