## [Decision Letter · Decision Letter 0]

13 Jul 2020

PONE-D-20-14173

Amianthoid transformation of costal cartilage matrix in children with pectus excavatum and pectus carinatum

PLOS ONE

Dear Dr. Fayzullin,

Thank you for submitting your manuscript to PLOS ONE. After careful consideration, we feel that it has merit but does not fully meet PLOS ONE’s publication criteria as it currently stands. Therefore, we invite you to submit a revised version of the manuscript that addresses the points raised during the review process.

We look forward to receiving your revised manuscript.

Kind regards,

Lin Han

Academic Editor

PLOS ONE

Journal Requirements:

2. Thank you for including the following funding information within the acknowledgements section of your manuscript; "The study was sponsored by Russian academic excellence project "5-100"; and the Russian Science Foundation (Grant № 14-25-00055)."

"The authors received no specific funding for this work."

Reviewers' comments:

Reviewer's Responses to Questions

**Comments to the Author**

1. Is the manuscript technically sound, and do the data support the conclusions?

Reviewer #1: Partly

Reviewer #2: Yes

2. Has the statistical analysis been performed appropriately and rigorously? 

Reviewer #1: No

Reviewer #2: N/A

3. Have the authors made all data underlying the findings in their manuscript fully available?

Reviewer #1: Yes

Reviewer #2: Yes

4. Is the manuscript presented in an intelligible fashion and written in standard English?

Reviewer #1: Yes

Reviewer #2: Yes

5. Review Comments to the Author

Reviewer #1: The study characterized the relationship between amianthoid transformation and pectus excavatum/carinatum in children popolation. The concept of the study is novel and the background literature is thorough and clear. The experimental design is rigorous and appropriate and results show support of the conclusion. The study has broader impact on understanding the formation of pectus excavatum/carinatum and possibly develop strategies in prevention and/or treatment.

Comments:

1. As a general suggestion, images in figures 1-5, as well as the corresponding supplemental figures, are not ordered in a directly comparison manner. This makes the figures difficult to read and weakens the support of the figures towards the conclusion. It's recommended to rearrange the images to represent the comparison between the control and PE/PC groups, for each type of AT.

2. Scale bars of the images 1-5 are inconsistent. Images with significantly different scale bars can give false outcome due to the contrast between different regions in view, for example figure parts 1f,2b,4d. It's recommended to adjust to or replace with images with similar scale bars for direct comparison.

3. Figure 5 seems to be missing several components. The SEM part contains the canonical and find-fibred types for control group, and other types for PE group. The AFM part contains fine fibred type for control group, and other types for PC group. The purpose of presenting this figure is unclear as it doesn't show any direct comparison between experimental groups for each type of AT.

4. In the fibril diameter measurement of the TEM images, measurements were taken with minimal, medium and maximum diameters of 10 randomly selected fibrils. The definition is unclear and might not properly represent the fibril thickness distribution of corresponding sample/region. It's recommended to have at least 50-100 measurements of randomly selected fibrils and have the distributions plotted (for example, a histogram) and compared.

5. It's not clearly stated which statistical test is implemented for Figures 6 and 7. From my understanding Figure 6 represents the results from the two-way anova with Tukey posthoc test and Figure 7 represents the results from the Spearman's correlation test. However Figure 6 only showed the comparison between different types of AT, but not the comparison between experimental groups. Also it's not described how the family-wise error was addressed in the two-way anova. In the methods section it's described that the correlations between the patients' age and the mean area of different types of AT areas was examined using the Spearman's correlation test, but result is not shown.

Also a grammatical error in the abstract line 42 needs to be corrected: "AT areas of were identified..."

Reviewer #2: The authors presented an interesting study that the foci of amianthoid transformation was found in costal cartilage extracellular matrix for both normal and pathological conditions. In addition, by using multiple imaging techniques, authors identified four types of AT areas, which occurrence frequency was associated with costal cartilage disorder. This discovery could potentially serve as an important criteria for future study of pectus excavatum and pectus carinatum. However, there are several issues to be addressed before publishcation,

1. Since both pectus excavatum and pectus carinatum were found to have a higher incidence rate in male, I’m interested in whether gender is a factor to affect the AT patterns, especially in abnormal conditions?

2. In Figure 6, how did author determine the minimal, medium and maximum fibril diameter? I also would like to look at the representative images of minimal, medium and maximum fibrils in control, PE and PC conditions.

3. In Figure 2, it is confusing that collagen I and III IHC (2a, f) were stained in the similar region and imagined under same scale bar, while no such scale bar image was provided for collagen II staining, and cellularity in col II stained images seemed to be lower. At least one similar region under 100 µm scale bar image of col II IHC should be provided for better comparison.

4. Line 315 Table 1, what is the criteria to define the IHC staining intensity? Is that based on color threshold or subjective observation?

5. Line 313, why it is “surprisingly” to not observe a negative staining of collagen I and III in a collagen II dominated cartilage tissue?

6. PLOS authors have the option to publish the peer review history of their article (what does this mean?). If published, this will include your full peer review and any attached files.

Reviewer #1: No

Reviewer #2: No

---

## [Author Response · Author response to Decision Letter 0]

27 Nov 2020

23.11.2020

Dear Lin Han,

Thank you very much for providing us an opportunity to revise our manuscript and to address the reviewers’ comments. 

We wish to thank the reviewers for their time and valuable comments, which were constructive and helpful to improve the quality of our work. Our responses to the reviewers’ comments are listed below. 

All corrections made according to the reviewer’s comments are included into the manuscript. The changes are highlighted in a marked-up copy of the manuscript. 

We hope the revised version of our manuscript fits the PLOS One requirements. We appreciate your consideration of our work and are looking forward to hearing from you.

Best regards,

Alexey Fayzullin, MD, the corresponding author.

To editor: 

Answer: 

We checked the revised manuscript for style errors and named the uploaded files as required.

2) Thank you for including the following funding information within the acknowledgements section of your manuscript; "The study was sponsored by Russian academic excellence project "5-100"; and the Russian Science Foundation (Grant № 14-25-00055)."

"The authors received no specific funding for this work."

Answer: 

The mentioned sources did not have any specific connections to the study nor they were spent on the researchers’ salaries for the project. However, the experimental work was conducted on the equipment bought with these funds for other projects. Since there is no direct connection to the study or the researchers we propose to erase the mentioning of these sources. We also amended the text of the Acknowledgment section accordingly.

3) We note that you have included the phrase “data not shown” in your manuscript. Unfortunately, this does not meet our data sharing requirements. PLOS does not permit references to inaccessible data. We require that authors provide all relevant data within the paper, Supporting Information files, or in an acceptable, public repository. Please add a citation to support this phrase or upload the data that corresponds with these findings to a stable repository (such as Figshare or Dryad) and provide and URLs, DOIs, or accession numbers that may be used to access these data. Or, if the data are not a core part of the research being presented in your study, we ask that you remove the phrase that refers to these data.

Answer: 

We added the relevant figure panel as S12 Figure.

To reviewer 1: 

1) As a general suggestion, images in figures 1-5, as well as the corresponding supplemental figures, are not ordered in a directly comparison manner. This makes the figures difficult to read and weakens the support of the figures towards the conclusion. It's recommended to rearrange the images to represent the comparison between the control and PE/PC groups, for each type of AT..

Answer: 

We added new organized figure panels (see Figs 1-10).

2) Scale bars of the images 1-5 are inconsistent. Images with significantly different scale bars can give false outcome due to the contrast between different regions in view, for example figure parts 1f,2b,4d. It's recommended to adjust to or replace with images with similar scale bars for direct comparison.

Answer:

We added new panels with better magnification co-localization of figures (see Figs 1-10).

3) Figure 5 seems to be missing several components. The SEM part contains the canonical and find-fibred types for control group, and other types for PE group. The AFM part contains fine fibred type for control group, and other types for PC group. The purpose of presenting this figure is unclear as it doesn't show any direct comparison between experimental groups for each type of AT.

Answer:

We added new panels with figures for each group and type of matrix (see Figs 9, 10).

4) In the fibril diameter measurement of the TEM images, measurements were taken with minimal, medium and maximum diameters of 10 randomly selected fibrils. The definition is unclear and might not properly represent the fibril thickness distribution of corresponding sample/region. It's recommended to have at least 50-100 measurements of randomly selected fibrils and have the distributions plotted (for example, a histogram) and compared.

Answer:

We changed our morphometry protocol. First, we made new TEM images at 12,000 magnification. This allowed us to randomly select 10 images for each type of matrix in each sample. Each selected TEM image was examined in detail by two experienced pathologists to evaluate the diameters of 50 selected fibrils and their aggregates. One aggregate with a maximum diameter and one fibril with a minimum diameter were chosen in each image. The remaining 48 fibrils / aggregates in the image were selected randomly. 

We obtained 500 values of the diameters of fibrils and aggregates, including 10 values of the minimum diameters of fibrils and 10 values of the maximal diameters of aggregates in each group, for each type of matrix in each tissue sample. Subsequently, we calculated mean medium diameters, mean minimal diameters and mean maximal diameters for each sample.

We added the relevant text to the Methods section. We also added the required interleaved scatter graph with bars as S13 figure.

5) It's not clearly stated which statistical test is implemented for Figures 6 and 7. From my understanding Figure 6 represents the results from the two-way anova with Tukey posthoc test and Figure 7 represents the results from the Spearman's correlation test. However Figure 6 only showed the comparison between different types of AT, but not the comparison between experimental groups. Also it's not described how the family-wise error was addressed in the two-way anova. In the methods section it's described that the correlations between the patients' age and the mean area of different types of AT areas was examined using the Spearman's correlation test, but result is not shown.

Answer:

The test in mentioned figures (in revised version – Figs 11-13) was Two-way ANOVA followed by the Tukey's test. We added a new Figure with comparison between the study groups and made relevant changes to the text. The results of the Spearman’s correlation test are presented in S1 Table. 

6) Also a grammatical error in the abstract line 42 needs to be corrected: "AT areas of were identified...".

Answer:

The error was corrected.

To reviewer 2:

1) Since both pectus excavatum and pectus carinatum were found to have a higher incidence rate in male, I’m interested in whether gender is a factor to affect the AT patterns, especially in abnormal conditions?

Answer:

We believe that there must be a correlation between gender and AT patterns, especially in adolescents. In general, we hypothesize that these patterns represent the maturation of the costal cartilage and change over the course of life. However, the specific question can be systematically addressed in a new study with larger groups of patients standardized by both gender and age. 

2) In Figure 6, how did author determine the minimal, medium and maximum fibril diameter? I also would like to look at the representative images of minimal, medium and maximum fibrils in control, PE and PC conditions.

Answer:

The measurements were conducted in TEM images made at 12,000x magnification. We made 50 measurements in each image and determined minimal and maximal diameters for each image taken. Medium diameter was calculated from 50 measurements. You can find the representative images with red dash-dotted brackets in a revised Figure 8.

3) In Figure 2, it is confusing that collagen I and III IHC (2a, f) were stained in the similar region and imagined under same scale bar, while no such scale bar image was provided for collagen II staining, and cellularity in col II stained images seemed to be lower. At least one similar region under 100 µm scale bar image of col II IHC should be provided for better comparison.

Answer:

We completely revised the IHC panel and organized it better.

4) Line 315 Table 1, what is the criteria to define the IHC staining intensity? Is that based on color threshold or subjective observation?

Answer:

The IHC staining intensity was evaluated by the observation and semiquantitative scores due to the uneven expression of collagen type II in AT areas. The non-linear dependence of the IHC staining intensity on the local concentration of the targeted antigen and the semi-quantitative nature of the conventional method (Kashyap, A., A. Fomitcheva Khartchenko, P. Pati, M. Gabrani, P. Schraml and G. V. Kaigala (2019). "Quantitative microimmunohistochemistry for the grading of immunostains on tumour tissues." Nat Biomed Eng 3(6): 478-490) were also considered. We added the relevant sentence to the Methods section.

5) Line 313, why it is “surprisingly” to not observe a negative staining of collagen I and III in a collagen II dominated cartilage tissue?

Answer:

We rephrased this sentence in the results section. The amended phrase says: “Collagen types I and III, abnormal for the matrix of costal cartilage, were not observed either in the intact cartilage, or in the AT areas (Table 1, S10, S11 Figs).”

In the introduction section we wrote: “According to H. Claassen et al. [22], AFs in the thyroid cartilage contain collagens types II, IX and XI, resembling composition of the normal organ-specific ECM. Collagen type I, abnormal for the healthy hyaline cartilage, was found in AFs after puberty”.

---

## [Decision Letter · Decision Letter 1]

23 Dec 2020

Amianthoid transformation of costal cartilage matrix in children with pectus excavatum and pectus carinatum

PONE-D-20-14173R1

Dear Dr. Fayzullin,

We’re pleased to inform you that your manuscript has been judged scientifically suitable for publication and will be formally accepted for publication once it meets all outstanding technical requirements.

Kind regards,

Lin Han

Academic Editor

PLOS ONE

Additional Editor Comments (optional):

Reviewers' comments:

Reviewer's Responses to Questions

**Comments to the Author**

1. If the authors have adequately addressed your comments raised in a previous round of review and you feel that this manuscript is now acceptable for publication, you may indicate that here to bypass the “Comments to the Author” section, enter your conflict of interest statement in the “Confidential to Editor” section, and submit your "Accept" recommendation.

Reviewer #1: All comments have been addressed

Reviewer #2: All comments have been addressed

2. Is the manuscript technically sound, and do the data support the conclusions?

Reviewer #1: Yes

Reviewer #2: Yes

3. Has the statistical analysis been performed appropriately and rigorously? 

Reviewer #1: Yes

Reviewer #2: Yes

4. Have the authors made all data underlying the findings in their manuscript fully available?

Reviewer #1: Yes

Reviewer #2: Yes

5. Is the manuscript presented in an intelligible fashion and written in standard English?

Reviewer #1: Yes

Reviewer #2: Yes

6. Review Comments to the Author

Reviewer #1: Great work! All my previous comments have been addressed properly. I would suggest to remove the 400X magnification images from Figures 1-5 and possibly combine them into 1 or 2 figures for better comparison but that's optional.

Reviewer #2: All my comments have been addressed properly, and I would recommend this paper ready to be published.

7. PLOS authors have the option to publish the peer review history of their article (what does this mean?). If published, this will include your full peer review and any attached files.

Reviewer #1: No

Reviewer #2: No

---

## [Editor Report · Acceptance letter]

15 Jan 2021

PONE-D-20-14173R1 

Amianthoid transformation of costal cartilage matrix in children with pectus excavatum and pectus carinatum 

Dear Dr. Fayzullin:

I'm pleased to inform you that your manuscript has been deemed suitable for publication in PLOS ONE. Congratulations! Your manuscript is now with our production department. 

Kind regards, 

on behalf of

Dr. Lin Han 

Academic Editor

PLOS ONE